# Nutritional Intervention for the Elderly during Chemotherapy: A Systematic Review

**DOI:** 10.3390/cancers16162809

**Published:** 2024-08-09

**Authors:** Roberta Vella, Erica Pizzocaro, Elisa Bannone, Paola Gualtieri, Giulia Frank, Alessandro Giardino, Isabella Frigerio, Davide Pastorelli, Salvatore Gruttadauria, Gloria Mazzali, Laura di Renzo, Giovanni Butturini

**Affiliations:** 1Department of Hepato-Pancreato-Biliary Unit, Pederzoli Hospital, 37019 Peschiera del Garda, Italy; roberta.vella02@community.unipa.it (R.V.); elisa.bannone@ospedalepederzoli.it (E.B.); alessandro.giardino@ospedalepederzoli.it (A.G.); isabella.frigerio@ospedalepederzoli.it (I.F.); butturinichirurgo@gmail.com (G.B.); 2Department of Precision Medicine in the Medical, Surgical and Critical Care Area, University of Palermo, 90127 Palermo, Italy; 3PhD School of Applied Medical-Surgical Sciences, University of Tor Vergata, 00133 Rome, Italy; giulia.frank@students.uniroma2.eu; 4Section of Clinical Nutrition and Nutrigenomic, Department of Biomedicine and Prevention, University of Rome Tor Vergata, Via Montpellier 1, 00133 Rome, Italy; paola.gualtieri@uniroma2.it (P.G.); laura.di.renzo@uniroma2.it (L.d.R.); 5Collegium Medicum, University of Social Sciences, 90-113 Łodz, Poland; 6Department of Oncology Unit, Pederzoli Hospital, 37018 Peschiera del Garda, Italy; davide.pastorelli@ospedalepederzoli.it; 7Department for the Treatment and the Study of Abdominal Diseases and Abdominal Transplantation, IRCCS-ISMETT, University of Pittsburgh Medical Center Italy, 90127 Palermo, Italy; sgruttadauria@ismett.edu; 8Department of General Surgery and Medical-Surgical Specialties, University of Catania, 95123 Catania, Italy; 9Department of Medicine, Geriatrics Division, University of Verona, 37134 Verona, Italy; gloria.mazzali@univr.it

**Keywords:** systematic review, nutrition therapy, nutritional status, oral nutritional supplements, chemotherapy, older patients

## Abstract

**Simple Summary:**

Elderly cancer patients represent a population particularly susceptible to nutritional alterations, which can have an impact on overall survival and chemotherapy tolerance. The effects of nutritional interventions in this frail population have not been adequately examined in previous review studies. We conducted a systematic review of the existing literature on the effects of oral nutritional supplements (ONSs) and dietary counseling during chemotherapy in older oncology patients. Various types of ONS were investigated, including multimodal intervention with tailored nutritional counseling, whey protein supplements, amino acids supplements (including immune nutrition supplements), and fish oil omega-3-enriched supplements. ONSs showed promise in reducing chemotherapy side-effects and improving nutritional status in older cancer patients, but further studies are needed to explore their efficacy on chemotherapy adherence and overall survival. Further studies are needed to investigate the effects of ONS considering chronological age and frailty criteria, different dietary habits, and specific nutritional assessment like Bioelectrical Impedance Analysis.

**Abstract:**

This study aims to review existing literature on the effect of oral nutritional supplements (ONSs) during chemotherapy in older cancer patients. Electronic databases were searched for relevant studies up to March 2024. The risk of bias in the included studies was evaluated using the Cochrane tool. Eligible studies included randomized, prospective, and retrospective studies evaluating the effect of ONSs in elderly (median age > 65 years) cancer patients during chemotherapy. Data regarding chemotherapy adherence, toxicity, overall survival, and nutritional status were extracted. A total of ten studies, involving 1123 patients, were included. A meta-analysis of the results was not conducted due to the scarcity and heterogeneity of results. Some ONSs were associated with reduced incidence of chemotherapy side-effects, particularly oral mucositis, and improved nutritional status. There was limited or no evidence regarding the impact of ONSs on chemotherapy adherence or overall survival. Various types of ONS were investigated, including multimodal intervention with tailored nutritional counseling, whey protein supplements, amino acids supplements (including immune nutrition supplements), and fish oil omega-3-enriched supplements. ONSs showed promise in reducing chemotherapy side-effects and improving nutritional status in older cancer patients, but further studies are needed to explore their efficacy on chemotherapy adherence and overall survival. Future research should consider both chronological age and frailty criteria, account for dietary habits, and use specific nutritional assessment like Bioelectrical Impedance Analysis.

## 1. Introduction

Cancer is one of the leading causes of morbidity and mortality worldwide, with aging being a major risk factor. Indeed, the highest probability of developing invasive cancer occurs between the ages of 65 and 84 years [1]. 

Nutritional alterations are common in patients with cancer [2,3] due to the activation of systemic hypercatabolic and inflammatory states. Additionally, common chemotherapy side-effects, such as anorexia, nausea, and vomiting, further contribute to these nutritional alterations, leading to detrimental changes in metabolism and resulting in malnutrition [4,5,6]. Prolonged malnutrition can lead to cachexia, a multifactorial syndrome characterized by severe loss of body weight, as well as adipose and muscle tissue, with increased protein catabolism. The rate of cachexia in cancer patients is estimated to be as high as 62% [7,8], and lean tissue wasting due to cancer progression exacerbates treatment-related toxicities in patients receiving various chemotherapeutic agents [9]. This deterioration often forces cancer patients to interrupt chemotherapy, creating a vicious cycle that frequently leads to poor outcomes. 

In this context, nutritional interventions have been proposed to reduce chemotherapy-induced toxicity and enhance chemotherapy adherence. However, most studies investigating nutritional interventions in cancer patients have rarely considered age or used it to stratify their effects. Older patients represent a critical population because the mechanisms altered in cancer patients are further emphasized in older individuals [10]. The metabolism of elderly patients is shifted towards a catabolic state and chronic degenerative diseases often coexist, making older adults more vulnerable [11].

This systematic review examines the existing literature on the effect of oral nutrition therapy, in the form of oral nutritional supplements (ONSs) and dietary counseling, during chemotherapy in older oncology patients. The aim is to identify the benefits of ONS therapies and understand the reasons for their failure, thereby providing a rationale for further investigation.

## 2. Materials and Methods

### 2.1. Information Sources and Search Strategy

An electronic, systematic, and comprehensive literature review was conducted and reported following the PRISMA 2020 guidelines and AMSTAR 2 (Assessing the methodological quality of systematic reviews) guidelines [12,13]. The search covered all of the relevant literature published from 2014 through March 2024 using MEDLINE (Pubmed), Scopus, Google scholar, and Embase. References from included studies were also checked to identify any additional relevant papers. The following search terms were used: “Nutritional interventions”, “Diet”, “Chemotherapy”, “Elderly”. The full search strategy for PubMed is detailed in Appendix B. The study protocol was registered on PROSPERO (ID: CRD42024519312).

### 2.2. Eligibility Criteria

Randomized controlled trials (RCTs) and prospective and retrospective studies assessing the effects of nutritional interventions in elderly cancer patients undergoing chemotherapy were included. Studies reporting a mean or median age over 65 years for the study population were considered eligible. Intervention studies had to test an ONS, defined as an oral nutritional therapy in the form of a single- or multi-nutrient supplement or a system to support nutritional intake (e.g., dietary counseling) compared with usual care or an alternative intervention (e.g., placebo). Interventions designed to optimize nutritional status via enteral or parenteral nutrition, chemical additives, or medications (e.g., steroids to stimulate appetite) were excluded. Studies involving ONS administration outside the context of chemotherapy, non-English papers, case reports, and studies not involving humans were also excluded.

### 2.3. Selection and Data Collection Processes

The screening process was performed using the free online app Rayyan (http://rayyan.qcri.org accessed on 17 February 2024). After removing duplicates, two authors (RV and EP) independently screened titles and abstracts to identify potentially eligible studies. The full texts of the selected studies were retrieved and independently assessed for eligibility by the same authors. Any uncertainties regarding the inclusion of the studies were resolved by consensus and, if necessary, by consulting a third author (GB). Appendix C displays the list of reports excluded.

### 2.4. Data Items

Data were extracted into an Excel sheet (Microsoft Excel Version 17, Microsoft Corporation 19) and analyzed using Review Manager (Revman) version 5.4 (The Cochrane Collaboration, available at revman.cochrane.org). Extracted data included the following: author, year of publication, study design, location of the study, trial registration number, funding or sponsorship, blinding, number of included patients, study aim, inclusion and exclusion criteria, cancer site, and specifics of the nutritional intervention. Demographic data (i.e., age, gender) and body mass index (BMI) [14] were also obtained as mean (with standard deviation) or median (with interquartile range) for continuous variables, and numbers with percentages for categorical variables. 

### 2.5. Outcomes Measured

#### 2.5.1. Primary Outcomes

The following outcome measures were retrieved if reported by each study:ONS-related chemotherapy adherence: Defined as the completion of chemotherapy treatments according to the study protocol.ONS-related chemotherapy toxicity: Measured by the incidence of chemotherapy side effects.ONS-related overall survival: Defined as the time from chemotherapy initiation to the date of death from any cause or the last follow-up visit.

For these variables, the frequency of the effect was retrieved, and when available, the measure of the effect was also reported as Risk Ratio (RR) with a 95% confidence interval (CI).

#### 2.5.2. Secondary Outcome

ONS-related nutritional status: evaluated through anthropometric measurements (e.g., body weight gain and BMI) and/or body composition analysis (e.g., Bioelectrical Impedance Analysis—BIA or radiologic imaging). The prevalence of cachexia, defined as weight loss greater than 5% or weight loss greater than 2% in individuals already showing depletion [15], was also retrieved.

### 2.6. Study Risk of Bias Assessment

The risk of bias was assessed using the Cochrane Risk of Bias Assessment tool 2 (Cochrane collaboration, 2019) for randomized trials, and tabulated using the ROBVIS tool [16]. The assessment considered five domains: sequence generation, allocation concealment, blinding, incomplete outcome data, selective outcome reporting, and other potential sources of systematic bias. For each study, the risk of bias was ranked as low, high, or with some concerns. 

Primary and secondary outcomes were reported by only a few papers, and due to the heterogeneity in both measures of effect and types of ONS, a meta-analysis was not conducted. Therefore, a systematic narrative presentation of the study results is provided.

## 3. Results

A total of ten studies, encompassing 1123 patients, were included (Figure 1). The characteristics of the included studies are reported in Table 1. Eight studies focused on patients with gastrointestinal cancer [17,18,19,20,21,22,23,24], while the remaining two included patients with mixed type of cancers (i.e., gastrointestinal cancer, lung cancer, breast cancer, and lymphoma) [25,26]. There was a predominance of male patients (65%) compared to female patients. BMI data was available in seven studies [17,18,20,21,22,23,26], all reporting a normal BMI [14] in both ONS and control groups (Table 2).

### 3.1. Quality of Included Studies

Among the included studies, nine were RCTs [17,18,19,20,21,22,24,25,26], while one was a retrospective cohort study [23]. The overall quality of the studies was judged to have some concerns, with three studies assessed as having a low risk of bias [22,24,25]. The main sources of bias included the lack of blinding of both participants and outcomes assessors and the absence of appropriate analysis to account for missing data (Figure 2). Two RCTs [22,24] were funded by companies involved in the production of the ONS used in the trials (Appendix A).

### 3.2. Primary Outcomes: Chemotherapy Adherence, Chemotherapy Toxicity, and Overall Survival

The results of primary outcomes are displayed in Table 3. Data on ONS-related chemotherapy adherence were limited to two studies [20,22]. Although the adherence rates were similar between the two studies, around 80% in both the ONS and control groups, the studies varied in terms of chemotherapy duration (two [20] and six cycles [22], respectively) and the reasons for chemotherapy discontinuation (oral mucositis in the study of Tanaka et al. [20], and both hematological and non-hematological toxicities in the study of Khemissa et al. [22]).

Almost all studies reported the frequency of ONS-related chemotherapy toxicity in both ONS and control groups [17,19,20,21,22,24,25,26], but quantitative measures of the ONS effect were available for only two studies [20,26]. Most of the studies reported a reduced incidence of chemotherapy toxicity with ONS [19,20,21,22,24,25,26], particularly gastrointestinal side-effects [19,20,21,24]. 

The definition of chemotherapy toxicity varied among studies, with four studies specifically analyzing the incidence of oral mucositis [19,20,21,24], while the others [17,22,25,26] considered the overall incidence of any side effects during chemotherapy treatment (Table 3). 

None of the included studies evaluated the effect of ONS on overall survival (Table 3).

### 3.3. Secondary Outcome: Nutritional Status

Seven studies analyzed the effect of nutritional intervention on patients’ nutritional status. All studies defined the nutritional status using anthropometric measurements (i.e., weight gain and BMI) [17,18,20,23,24,25,26], reporting mixed results. Three studies reported a significant difference in weight gain [18,20,24] in favor of ONSs compared to control groups, whereas two studies found no difference [17,25].

Only two studies assessed the effect of ONSs on nutritional status using BIA [18,26], showing more consistent results. Both studies demonstrated a significant difference in favor of the ONS group compared to controls, with one study [26] analyzing the phase angle and the other examining appendicular skeletal muscle mass [18]. None of the studies specifically investigated the effect of ONS on nutritional status using radiologic imaging, such as the psoas muscle index measured by computed tomography. 

Only one study [18] specifically reported the rate of cachectic patients [18], which was 41.6% of elderly cancer patients.

### 3.4. Types of Nutritional Intervention

The included studies evaluated various ONSs (Table 1):Multimodal interventions [25];Whey protein supplements [26];Amino acid supplements [17,18,19,20,21,22,24];Fish oil omega-3-enriched supplements [23].

#### 3.4.1. Multimodal Interventions 

One study evaluated the effect of a multimodal intervention that included tailored nutritional counseling [25]. The results showed that individual dietary counseling was associated with an increase in dietary intake but did not result in a decrease in mortality rates at one (RR = 1.1, 95%CI = 0.8–1.5; *p* = 0.74) and two years (RR = 1.1, 95%CI = 0.9–1.5; *p* = 0.37). The daily caloric and protein targets (30 kcal/kg/d and 1.2 g protein/day, respectively) were not achieved in either the control or in the multimodal intervention group. The rate of patients requiring amino acid supplementation was higher in the control group compared to the multimodal intervention group.

#### 3.4.2. Whey Protein Supplements 

One parallel-group RCT [26] compared whey protein supplementation, consisting of the soluble class of dairy proteins, with placebo in elderly cancer patients during chemotherapy. The whey protein supplementation group showed lower chemotherapy toxicity (*p*= 0.009) and improved nutritional status, with a significant increase in fat-free mass, body weight, and phase angle (measured with BIA) (*p*= 0.041, *p*= 0.023, *p* = 0.027, respectively) compared to the placebo group. 

#### 3.4.3. Amino Acid Supplements

Seven studies evaluated the effect of amino acid supplementation [16,17,18,19,20,21,23], including single amino acids [18,20], mixtures of amino acids [17,19,20,21,24], and combinations with transforming growth factor-beta 2 (considered immune nutritional supplements) [22]. 

The studies on single amino acid supplements investigated L-leucine [18] and glutamine [20]. L-leucine [18] supplementation improved nutritional status, showing a significant increase in body weight after 8 weeks of treatment (*p* = 0.01) compared to a stable body weight (*p* = 0.85) in the placebo group. L-leucine also significantly increased appendicular skeletal muscle mass, even in cachectic patients. Glutamine supplementation was not associated with reduced incidence of high-grade oral mucositis in patients with esophageal cancer undergoing chemotherapy [20] (seven cases in the intervention group and six cases in the control group, *p* > 0.05).

Five studies investigated ONSs containing a mixture of amino acids [17,19,20,21,24]. All the studies used an elemental diet powder, composed of a mixture of essential amino acids (valine, isoleucine, leucine, methionine, phenylalanine, tryptophan, threonine, histidine, and lysine) with non-essential amino acids (arginine, glycine, glutamine, proline, and tyrosine) and additives. The studies by Tanaka et al. [20,24], Okada et al. [21], and Toyomasu et al. [19] assessed the impact of amino acid mixtures on preventing oral mucositis in patients with esophageal [20,21,24], and gastric [19] cancer. These studies reported reductions in high-grade oral mucositis, higher chemotherapy adherence, and improvements in nutritional status, with significant weight gain [20] and reduced body weight loss [19] (all *p* < 0.05) in the mixture of amino acids groups compared to the control groups. Conversely, the RCT by Katada et al. [17] found no differences in overall clinically relevant gastrointestinal toxicity (*p* = 0.267) and in severe chemotherapy adverse events (all *p* > 0.05). Moreover, nutritional status after chemotherapy showed no significant difference between the two groups, in terms of both body weight (*p* = 0.057) and muscle mass (*p* = 0.056). The duration of supplementation varied between the included studies from 14 days [21] to 8–9 weeks [17,24] and was unreported in the other two studies [19,20].

The multicenter double-blind RCT by Khemissa et al. [22] investigated oral immune nutrition supplements containing glutamine and transforming growth factor-beta 2. There was no significant difference in the rate of non-hematological and hematological grade 3–4 toxicities between ONSs and placebo (*p* = 0.82 and *p* = 0.25, respectively). Anorexia was significantly higher in the ONS group than the control group (*p* = 0.02). 

#### 3.4.4. Fish Oil Omega-3-Enriched Supplements

The study by Shirai et al. [23] analyzed the effect of fish-oil-enriched ONSs by randomly assigning gastrointestinal cancer patients undergoing chemotherapy to receive one/two packs of fish-oil-enriched ONSs daily for six months, based on their treatment tolerance versus standard care. The study found that fish-oil-enriched nutrition was associated with greater chemotherapy tolerance, and improved nutritional status, as evaluated by BIA (Table 3). Increased chemotherapy adherence was particularly evident in patients with systemic inflammation, as defined by a Glasgow Prognostic Score [27] of 1 or 2.

## 4. Discussion

This systematic review suggests some evidence supporting the effectiveness of ONSs in reducing chemotherapy toxicity and enhancing nutritional status among elderly cancer patients undergoing chemotherapy. However, there is insufficient evidence regarding the impact of ONSs on chemotherapy adherence or survival in this frail population. The studies included in this review exhibited extreme heterogeneity in terms of type, schedule, and duration of ONS treatment, which precludes definitive conclusions.

Cachexia and malnutrition are prevalent among elderly cancer patients undergoing chemotherapy, with rates as high as 62% [7]. These conditions are influenced by different factors. Chemotherapy exacerbates nutritional decline by increasing catabolism and impairing dietary intake due to side effects such as nausea, vomiting, loss of appetite, and taste changes [28]. These mechanisms are compounded in elderly patients, who already experienced hormonal changes and immune alterations [29], leading to increased proteolysis and reduced anabolism [30]. Recognizing these syndromes is crucial, as studies have shown that one-year mortality doubles in older cancer patients undergoing chemotherapy who are malnourished or at risk of malnutrition [31]. In this context, effective nutritional interventions have the potential to influence treatment outcomes, quality of life, and overall survival [32].

ONSs have shown promising results in reducing chemotherapy toxicity and improving nutritional status in cancer patients [33,34], but evidence supporting their efficacy in elderly patients is less robust.

Interestingly, studies that reported on chemotherapy adherence in elderly patients found similar rates between ONS and control groups [20,22,23], despite ONSs being often associated with overall reductions in chemotherapy toxicity [20,21,22,24,25,26]. These inconsistencies may stem from variations in outcome measures used across studies, reflecting differences in type of primary tumor and polychemotherapy regimens applied. Additionally, the diversity in types of ONS used among studies further complicates the interpretation and the strength of the evidence. While promising results have been observed in the broader cancer population [35,36], no studies have investigated the impact of ONSs on overall survival specifically in elderly patients. Indeed, older patients face additional challenges compared to the general population, such as polypharmacy, comorbidities, and age-related physiological changes like social isolation, cognitive impairment, and depression. These factors can significantly impact their ability to adhere to treatment protocols and cope with treatment-related side effects, affecting compliance and dropout rates, and remain inadequately addressed in the literature [37,38,39,40].

While this systematic review did not find consistent evidence supporting the widespread use of ONSs in elderly cancer patients, it underscored the diverse array of ONS types available for potential therapeutic use and the presumed mechanisms involved (Figure 3).

The simplest nutritional support to prevent alterations in nutritional status during chemotherapy in elderly cancer patients could be a multimodal intervention that includes personalized dietary counseling. Although integrating dietary counseling into the oncological care pathway increased oral intake in elderly patients undergoing chemotherapy [25], many did not achieve their daily caloric and protein targets, necessitating additional ONSs, albeit in a reduced percentage compared to the control group. These findings raise questions about the efficacy of dietary counseling alone without additional ONSs. It is important to note that patients’ baseline dietary habits were not reported and could have varied across studies conducted in different countries. A healthy eating pattern could diminish the apparent effectiveness of dietary counseling by reducing differences between treatment and control groups. The only study addressing multimodal treatment in elderly patients was conducted in the Mediterranean area [25], indicating the need for further research in other regions to account for variations in standard dietary habits.

Ensuring an adequate protein intake in cancer patients during chemotherapy represents a cornerstone of dietary therapy [41,42]. The European society for clinical nutrition and metabolism (ESPEN) guidelines recommend a minimum protein intake of 1.0 g/kg/day and up to 1.5 g/kg/day for cancer patients to help maintain or restore lean body mass [43].

Despite these guidelines, only one trial [26] in this review specifically investigated whey protein ONSs, highlighting its positive effect in frail, elderly, malnourished patients by reducing chemotherapy toxicity and improving nutritional status. Indeed, daily protein intake was satisfactory only in the whey protein ONS group, consistent with ESPEN recommendations [43], thereby helping to trigger a virtuous circle with increased muscle protein stores, improved body composition, and reduced chemotherapy toxicity. 

Unlike protein-based ONSs, amino acid supplementation in cancer patients targets not only the restoration of nutritional status but also the reduction in systemic inflammation and the prevention of chemotherapy-induced gastrointestinal side-effects through mechanisms such as immune modulation and metabolic regulation [44,45]. This review included studies investigating supplementation of single or mixtures of amino acid ONSs in elderly cancer patients. The outcomes of amino acid ONSs varied [17,19,20,21,24], although the majority indicated a reduced incidence of chemotherapy-induced oral mucositis in elderly cancer patients receiving amino acid ONSs [19,20,21,24]. The inconsistencies may be attributed to differences in supplement formulations, including variations in dosage, treatment duration, and patient compliance. Notably, ONSs of leucine, which enhances mitochondrial and protein metabolism via mTORC1 activation [44], and glutamine, which supports enteral mucosal integrity [45,46,47], as evidenced by elevated plasma diamine oxidase activity—a reliable marker of intestinal mucosal health [20]—were included. This effect is particularly significant in the prophylaxis of mucositis, a frequent complication associated with treatments like docetaxel, cisplatin, and 5-fluorouracil [24]. Interestingly, the administration of glutamine alone or in combination with transforming growth factor-beta 2 ("immuno-nutritional supplement") did not produce the same preventive effects on chemotherapy toxicity [22,24] as a mixture of amino acids comprising glutamine. This result is in contrast with previous studies on the adult oncological population [48,49] reporting the effectiveness of glutamine alone in preventing and treating oral mucositis in the oncological population through a local and immunomodulating effect. Indeed, glutamine represents the main energy source for cells of the immune system, and for the oral and intestinal epithelium; therefore, its supplementation is supposed to accelerate the healing of oral and intestinal mucosa [48]. This discrepancy might be due to a lower dosage used (13.5 g/day in the study of Khemissa et al. [18], compared to a range of 30–45 g/day reported in literature [48,49] as being effective) and lower availability of single amino acids in the elderly population; moreover, it suggests that a multi-amino-acid supplementation strategy might be more effective for nutrient delivery during chemotherapy in the elderly.

The plethora of supplements of single amino acids being studied in different clinical context is broad and encompasses amino acids with supposed anti-tumoral effects, such as arginine [50]. Despite some evidence regarding its efficacy for preventing chemotherapy toxicities, data on elderly patients are lacking. This gap underlines the need for studies addressing the issue of nutritional supplementation of both essential and non-essential amino acids in the elderly population, whose nutritional needs and pharmacokinetics are peculiar [51]. Finally, another controversial point is the inconsistency of results between studies evaluating single amino acids and those assessing a mixture of such. All studies conducted with a mixture of amino acids used the same formula of essential and non-essential amino acids, with different dosages and timing [17,19,20,21,24]. Despite most of the results [19,20,21,24] tending towards a reduction in side effects, especially oral mucositis, and increased chemotherapy adherence, it is worthy to note that the dosage of the mixture of amino acids used in these studies is equivalent to about one-third of the daily amino acid intake of healthy adults, suggesting that beyond amino acid concentrations, the set of amino acids probably exerts ancillary effects and future research is needed for understanding its effects on patients receiving anticancer therapy. The higher efficacy of amino acid mixtures [17,19,20,21,24] compared to single amino acids [18,20] suggests that a multi-amino-acid supplementation strategy might be more effective for nutrient delivery during chemotherapy in the elderly.

In addition to amino acids, lipid-derived supplements have also been investigated, demonstrating a positive effect of fish-oil-enriched ONSs in older patients with gastrointestinal cancer undergoing chemotherapy [23]. These benefits are likely due to the immune-modulating properties of fish oil, which include the formation of prostanoids and leukotrienes with low pro-inflammatory and immunosuppressive effects. 

This review has several limitations. The scarcity of results necessitated including articles with patients averaging over 65 years old, introducing age heterogeneity. Some studies focused on elderly patients specifically, while others included a broader age range. However, chronological age alone may not define elderly status [51] and future studies should incorporate additional biological aging parameters for better characterization. Since only a few papers [20,26] provided outcome measures of effect, and there was significant heterogeneity in the types of ONSs used, duration of administration, and chemotherapies, performing a meta-analysis was not feasible. Moreover, concerns about the quality of the included studies were noted, particularly due to the absence of blinding, pre-specified analysis plans, and issues with missing data, that limited conclusive conclusions on ONSs in elderly cancer patients. This might not be surprising, considering that concerns still exist on whether higher energy intake before chemotherapy is favorable. Some pre-clinical and clinical studies have shown a positive effect of low-calorie diets and short-term fasting on chemotherapy tolerance and effectiveness [20]. No studies analyzed the effects of intermittent fasting or ketogenic diets in elderly cancer patients, despite emerging evidence supporting these approaches [52,53]. The results of the ongoing BREAKFAST-2 trial will shed light on a fasting-like approach as a co-adjuvant in the multimodal treatment of cancer patients across a wide spectrum of tumors [54]. Similarly, interactions between dietary interventions and microbiome were not investigated. Future studies could provide insights into how environmental factors, such as diet, ONSs, and chemotherapy drugs, interact with host genetics, microbiome, and cancer cells. 

Although the extreme heterogeneity of the ONSs used in the included studies may be seen as a limitation, it also allowed for a broader exploration of nutritional interventions. This might highlight that no ONS is universally superior; rather, the most appropriate one should be chosen based on various factors, including the patient’s individual characteristics, primary tumor type, proposed chemotherapy treatment, and previous dietary habits. In this regard, it must be noticed that included studies were conducted in different locations (six out of ten studies included an Asian population [17,19,20,21,23,24]), with presumedly different environmental, lifestyle, dietary, and epigenetic factors. Despite the fact that interaction between dietary habits, environmental factors, and ONS effects as a potential bias was not explored, it underlies the multi-faceted nature of personalized nutritional interventions in elderly patients during chemotherapy. 

Finally, as regards the interaction between ONSs and nutritional status, only two studies [18,26] used quantitative and comprehensive analysis via BIA to assess the effect of ONS on nutritional status, while others [17,20,23,24,25] relied solely on surrogates such as weight gain or BMI. Among these, only one study [18] specifically analyzed the effect of ONSs and nutritional counseling in cachectic patients and reported a positive effect on skeletal muscle mass. This limitation may have obscured potential ONS effects and highlights the need for further studies to employ more comprehensive and specific measures for assessing nutritional status and a broader population, including both overweight and underweight patients. Such measures are crucial for effectively comparing different ONSs and understanding how they can affect energy intake and how their effects can vary according to patients’ nutritional status.

## 5. Conclusions

Maintaining skeletal muscle mass and physical function during chemotherapy in elderly cancer patients poses significant challenges due to cancer-related cachexia, chemotherapy-induced toxicities, and age-related malnutrition. The use of ONSs shows initial promise, particularly those containing whey protein, amino acids, or lipid supplements. These ONSs could reduce chemotherapy side-effects, such as oral mucositis, and enhance nutritional status. However, the evidence on the overall impact of ONS remains limited and fragmented. Further research focusing on elderly patients, considering both chronological age and frailty criteria, as well as reliable and reproducible methods for nutritional status evaluation, is essential to strengthen the evidence on the clinical effects of ONSs, personalized dietary counseling, and specific diets.

## Figures and Tables

**Figure 1 cancers-16-02809-f001:**
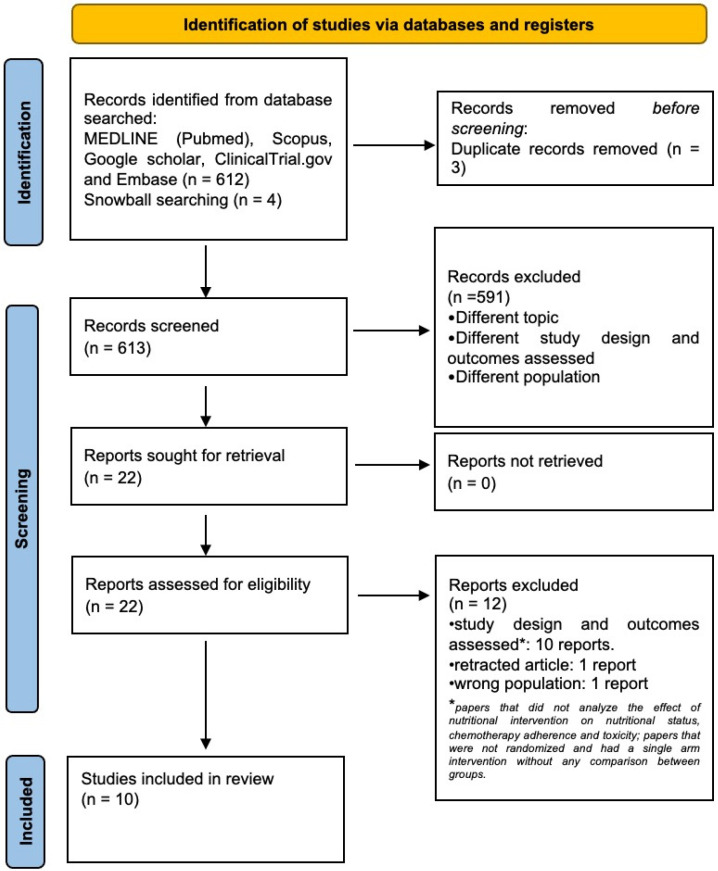
Flow diagram of the literature screening process and results.

**Figure 2 cancers-16-02809-f002:**
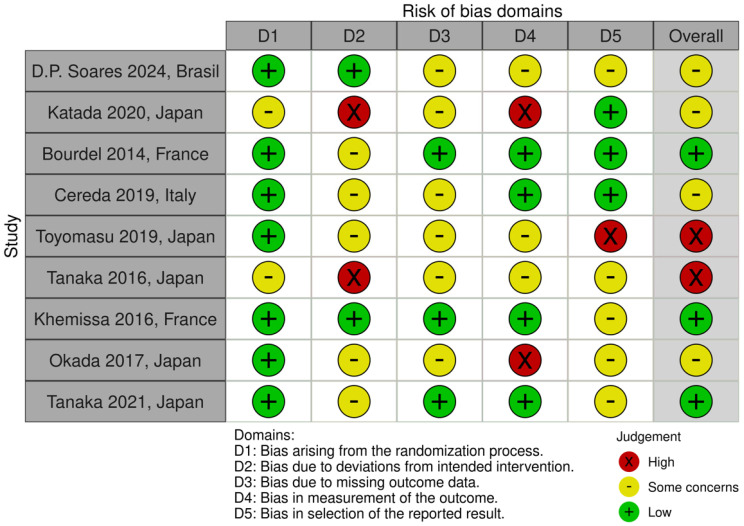
Quality assessment of included studies: D.P Soares et al. [18], Katada et al. [17], Bourdel – Marchasson et al. [25], Cereda et al. [26], Toyomasu et al. [19], Tanaka et al. 2016. [20], Khemissa et al. [22], Okada et al. [21], Tanaka et al. 2021 [24].

**Figure 3 cancers-16-02809-f003:**
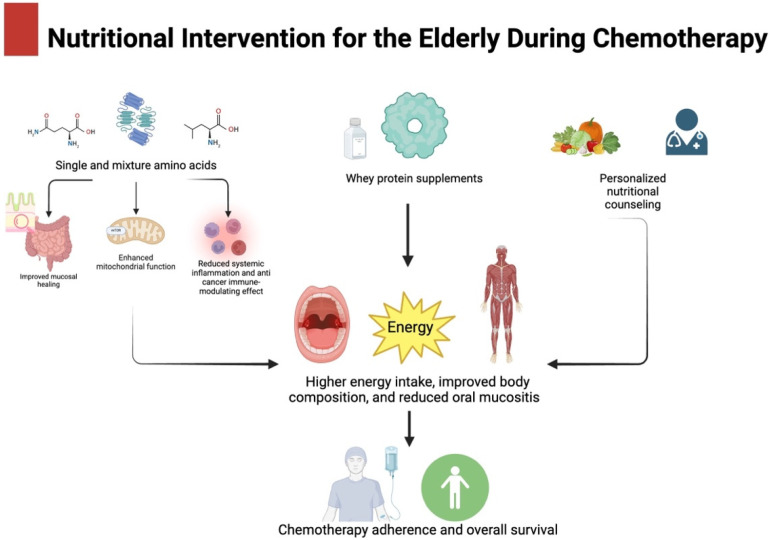
Mechanisms involved in beneficial effects of ONSs in elderly patients during chemotherapy. Figure created with BioRender.com.

**Table 1 cancers-16-02809-t001:** Characteristics of included studies. RCT: randomized controlled trials; TGF: transforming growth factor.

Author	Year	Location	Study Design	Aim	Inclusion Criteria	Exclusion Criteria	Intervention Group	Control Group
**Studies evaluating the effect of multimodal intervention**								
Bourdel-Marchasson et al. [25]	2014	France	RCT	Evaluate the effect of tailored nutritional counseling on mortality, toxicities, and chemotherapy outcomes	Patients older than 70 years with lymphoma or carcinoma and undergoing chemotherapy	Patients with cerebral metastasis and patients unable to take part in follow-up	Tailored nutritional counseling to achieve a daily protein and caloric goal of 1.2 g/kg/d and 30 kCal/kg body weight/d, respectively. Amino acid supplements were provided if pertinent	Usual dietary advice without daily protein and caloric intake goal
**Studies evaluating the effect of whey protein supplementation**								
Cereda et al. [26]	2019	Italy	RCT	Evaluate the effect of whey protein supplements on nutritional status and chemotherapy toxicity	Adult patients, malnourished (6-month unintentional weight loss ≥ 10%), and candidate to or undergoing chemotherapy	Patients aged <18 years and undergoing artificial nutrition (enteral or parenteral)	Whey protein supplementation in addition to usual nutritional counseling	Nutritional counseling without whey protein supplementation
**Studies evaluating the effect of amino acids supplementation**								
Katada et al. [17]	2020	Japan	RCT	Evaluate the effect of amino acid supplementation in form of “elemental diet” on chemotherapy toxicity	Adult patients with esophageal carcinoma, candidate to receive chemotherapy, and able to orally intake an elemental diet	Patients with a history of hypersensitivity to elemental diet	Mixture of amino acid supplements for 9 weeks after the start of chemotherapy	Patients in the control group did not receive any oral supplementation
D.P. Soares et al. [18]	2024	Brazil	RCT	Evaluate the effect of L-Leucine supplementation on nutritional status evaluated with body weight and body composition	Patients older than 60 years and candidates to receive chemotherapy	Patients younger than 60 years old, with mental disorders or cognitive or walking disabilities; patients undergoing artificial nutrition (enteral or parenteral)	Amino acid supplements (L-leucine) and nutritional counseling	Placebo
Toyomasu et al. [19]	2019	Japan	RCT	Evaluate the effect of amino acid supplementation on chemotherapy-induced oral mucositis or diarrhea	Adult patients with gastric cancer who had curative resection and are candidates to receive chemotherapy	Patients with severe heart disease, interstitial pneumonia or pulmonary fibrosis, bleeding tendency, liver cirrhosis or active hepatitis, chronic renal failure, severe diabetes and severe drug allergy	Mixture of amino acid supplements	Patients in the control group did not receive any oral supplementation
Tanaka et al., 2016 [20]	2016	Japan	RCT	Evaluate the effect of amino acid supplementation in form of “elemental diet” with glutamine for prevention of chemotherapy-induced oral mucositis	Adult patients with esophageal carcinoma and candidates to receive chemotherapy	Patients previously treated with chemotherapy for malignant disease or irradiation to major bone areas, patients with serious concomitant illness, symptomatic infectious disease, severe drug allergy, symptomatic peripheral neuropathy, or uncontrolled diabetes mellitus	Mixture of amino acid supplements with glutamine; glutamine supplements alone	Patients in the control group did not receive any oral supplementation
Khemissa et al. [22]	2016	France	RCT	Evaluate the effect of amino acid supplementation and transforming growth factor-beta 2 (the so-called “immune nutrition”) for chemotherapy-induced non-hematological toxicities	Adult patients with esophageal carcinoma and candidate to chemotherapy	Patients previously treated with chemotherapy for malignant disease or irradiation to major bone areas, patients with serious concomitant illness, symptomatic infectious disease, severe drug allergy, symptomatic peripheral neuropathy, or uncontrolled diabetes mellitus.	Amino acids supplements in form of “immunonutrition” containing TGF-beta and glutamine	Placebo
Okada et al. [21]	2017	Japan	RCT	Evaluate the effect of amino acid supplementation in form of “elemental diet” on chemotherapy-induced oral mucositis and diarrhea	Adult patients with esophagus carcinoma candidate to receive chemotherapy and without history of oral complications or immunodeficiency before chemotherapy	Patients simultaneously undergoing chemotherapy and radiation therapy	Amino acid supplements for 14 days and during chemotherapy	Patients in the control group did not receive any oral supplementation
Tanaka et al., 2021 [24]	2021	Japan	RCT	Evaluate the effect of amino acid supplementation in form of “elemental diet” on chemotherapy-induced oral mucositis	Adult patients with esophagus carcinoma and candidate to receive chemotherapy	Patients with symptomatic infectious disease, symptomatic peripheral neuropathy, patients with serious concomitant illness, patients with symptomatic bone or brain metastases, patients diagnosed with oral mucositis at registration	Mixture of amino acid supplements for 56 days during chemotherapy	Patients in the control group did not receive any oral supplementation
**Studies evaluating the effect of fish oil omega-3-enriched oral supplements**								
Shirai et al. [23]	2017	Japan	Retrospective cohort study	Evaluate the effect of fish oil omega-3-enriched oral supplements on chronological alterations in biochemical and physiological status during chemotherapeutic treatment	Adult patients with a clinical diagnosis of gastrointestinal cancer and candidate to receive chemotherapy	Not available	One or two packs of fish oil omega-3-enriched oral supplements per day for six months during chemotherapy	No additional nutritional treatment

**Table 2 cancers-16-02809-t002:** Demographic data, body mass index (BMI), and cancer site of population in included studies. NA: not available. Age is expressed in years as mean (standard deviation) or median (interquartile range); gender is expressed as prevalence of male patients (percentage); BMI: body mass index; BMI is expressed as mean (standard deviation) or median (interquartile range).

Author	Age (Years)		Gender Male (%)		BMI		Weight Gain		Cancer Site
	Control Group	Intervention Group	Control Group	Intervention Group	Control Group	Intervention Group	Control Group	Intervention Group	
**Studies evaluating the effect of multimodal intervention**									
Bourdel-Marchasson et al. [25]	78.3 ± 4.7	77.7 ± 5.2	91 (54.5)	81 (47.9)	NA	NA	4	5	Lymphoma or carcinoma (not specified)
**Studies evaluating the effect of whey protein supplementation**									
Cereda et al. [26]	65.7 ± 11.4	65.1 ± 11.7	49 (58.3)	47 (57.3)	22.3 ± 3.9	22 ± 4.1	0.7 (4.2)	1 (4.1)	Lung, stomach, esophagus, pancreas, colon, blood, breast, and head–neck cancer
**Studies evaluating the effect of amino acid supplementation**									
Katada et al. [17]	66.7 ± 5.0	67.8 ± 4.8	29 (82.9)	30 (83.3)	20.0 ± 2.9	20.7 ± 2.3	0.97 (5.4)	0.99 (3.9)	Esophageal cancer
D.P. Soares et al. [18].	65.00 ± 7.23	65.22 ± 8.19	18 (100%)	18 (100%)	22.41 ± 3.64	22.34 ± 2.79	0.22	2.27	Gastrointestinal and hepato-biliary-pancreatic cancer
Toyomasu et al. [19]	67.1 (59–80)	68.4 (61–80)	8 (73%)	9 (82%)	NA	NA	NA	NA	Gastric cancer
Tanaka et al. [20]	68 (49–82)	75 (58–83)	9 (90%)	10 (100%)	21.75 (18.37–25.59)	21.08 (14.60–24.20)	-5.40	1.70	Esophageal cancer
Khemissa et al. [22]	66 (60–75)	68 (61–74)	67 (66%)	63 (64%)	24.2 (22.0–27.2)	24 (21.0–26.1)	NA	NA	Gastrointestinal and hepato-biliary-pancreatic cancer
Okada et al. [21]	67.1	65.3	8 (80%)	9 (90%)	22.1	21.4	NA	NA	Esophageal cancer
Tanaka et al. [24]	68 (44, 86)	68 (34, 83)	50 (86)	43 (78)	NA	NA	NA	NA	Esophageal cancer
**Studies evaluating the effect of fish oil omega-3-enriched oral supplements**									
Shirai et al. [23]	68.9 ± 10.3	72.3 ± 8.4	64 (70%)	26 (70%)	21.7 ± 3.4	21.2 ± 2.9	NA	NA	Gastrointestinal and hepato-biliary-pancreatic cancer

**Table 3 cancers-16-02809-t003:** Outcomes measures in included studies. NA: not available; OR: odds ratio; IC: confidence interval.

Author	Chemotherapy Adherence					Chemotherapy Toxicity						Overall Survival				
	Control Group	Intervention Group	*OR*	*IC*	*p* Value	Control Group	Intervention Group	*OR*	*IC*	*p* Value	Chemotherapy Toxicity	Control Group	Intervention Group	*OR*	*IC*	*p* Value
**Studies evaluating the effect of multimodal intervention**																
Bourdel-Marchasson et al. [25]	NA	NA	NA	NA	NA	7 (10.4%)	7 (4.2%)	NA	NA	0.03	Infectious toxicities	NA	NA	NA	NA	NA
**Studies evaluating the effect of whey protein supplementation**																
Cereda et al. [26]	NA	NA	NA	NA	NA	83 (98.8%)	73 (89%)	−9.8	−16.9~−2.6	0.009	Any chemotherapy toxicity	NA	NA	NA	NA	NA
**Studies evaluating the effect of amino acid supplementation**																
Katada et al. [17]	NA	NA	NA	NA	NA	25 (71.4%)	30 (83.3%)	NA	NA	NA	Gastrointestinal toxicities	NA	NA	NA	NA	NA
D.P. Soares et al. [18]	NA	NA	NA	NA	NA	NA	NA	NA	NA	NA	NA	NA	NA	NA	NA	NA
Toyomasu et al. [19]	NA	NA	NA	NA	NA	3 (27.3%)	1 (9.1%)	NA	NA	NA	Gastrointestinal toxicities	NA	NA	NA	NA	NA
Tanaka et al. [20]	8 (80%)	8 (80%)	NA	NA	NA	6 (60%)	1 (10%)	0.1	0.0–0.6	0.02	Gastrointestinal toxicities	NA	NA	NA	NA	NA
Khemissa et al. [22]	88 (86%)	83 (84%)	NA	NA	NA	92 (90%)	88 (89%)	NA	NA	0.82	Any chemotherapy toxicity	NA	NA	NA	NA	NA
Okada et al. [21]	NA	NA	NA	NA	NA	8 (80%)	4 (40%)	NA	NA	0.02	Gastrointestinal toxicities	NA	NA	NA	NA	NA
Tanaka et al. [24]	NA	NA	NA	NA	NA	20 (34%)	8 (15%)	NA	NA	0.0141	Gastrointestinal toxicities	NA	NA	NA	NA	NA
**Studies evaluating the effect of fish oil omega-3-enriched oral supplements**																
Shirai et al. [23]	5	8	NA	NA	0.05	NA	NA	NA	NA	NA	NA	NA	NA	NA	NA	NA

## Data Availability

The dataset analyzed during the current study are not publicly available but are available from the corresponding author on reasonable request.

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
