# Peer review of "Nutritional Intervention for the Elderly during Chemotherapy: A Systematic Review"

_cancers, 2024, doi:10.3390/cancers16162809_

Round 1
Reviewer 1 Report
Comments and Suggestions for Authors
The paper “Nutritional Intervention for the Elderly During Chemotherapy: A Systematic Review” conducted a systematic review of the existing literature on the effects of oral nutritional supplements (ONS) and dietary counseling during chemotherapy in older oncology patients. The review is interesting and meaningful, but there are some questions that need to be further improved or explained.
Comments:
Q1. Please confirm whether ONS is a kind of therapy.
Q2. The types of cancer reported in these articles vary, the chemotherapy agents employed differ, the treatment methods utilized are diverse, and even the nutritional supplements administered exhibit variations. So how the authors obtained effective analysis results?
Q3. Some cited articles have reported ONS was amino acid oral liquid. What are the amino acid compositions and proportions of ONS in these papers? Are they consistent? It would also be interesting to determine which amino acids intake could reduce chemotherapy side effects.
Q4. Many studies have demonstrated the inhibitory role of arginine supplementation in tumor growth. However, it is puzzling why arginine is not widely available as a dietary supplement. Additionally, could you please elaborate on the advantages of the amino acids mentioned in this article? It would be beneficial if the author could further discuss the specific types of amino acids.
Q5. What is the distinction between protein intake and amino acids intake in the human body, particularly among older individuals? If we solely consider nutrition, would amino acid intake be more efficacious? When considering immune regulation, there are a plethora of substances available on the market with enhanced immune activity. Please enhance the expressions of relevant contents in disscussion section.
Q6. According to the content reflected in the article, it can be observed that the author has effectively organized the existing literatures; however, an original and comprehensive summary is lacking, particularly in the discussion section. It is recommended that the authors synthesize available literature data to formulate their own conclusive findings. If feasible, it is suggested that the authors could supplement a diagram showing the improvement mechanism of ONS on chemotherapy side effects.
Author Response
Dear Reviewer,
Please find attached the Word file with point-by-point response to all comments.

Reviewer 2 Report
Comments and Suggestions for Authors
This is a comprehensive review of nutritional intervention, focusing on oral nutritional supplements (ONS). Despite the limited number of publicly available articles, the authors conducted an excellent review using approximately 10 studies. The following comments may help improve the original manuscript:
Major Points
· Chemotherapy type, cancer type, chemo doses, frequencies, and other treatments such as radiation: Many factors could affect the impact of ONS. Please add more descriptions of these factors in the Discussion section.
· Analysis based on ~10 studies: Nine studies in Figure 2 include five studies from Japan. Please describe a potential bias in this review, since Asian/Japanese patients may not respond the same way as other populations.
· BMI – 21-24 in Table 2: The patients in this review generally have a relatively low BMI. This may be linked to the country of origin of the papers analyzed in this study. Please discuss the possible link between the value of ONS and BMI.
· Immune systems: It is recommended to discuss the possible effects of ONS on immune systems.
Minor Points
· Affiliation #1: #1 is used for two different institutions.
· Simple summary: The sentence starting with “The findings indicated …” should be proofread.
Comments on the Quality of English Languagea minor edit needed
Author Response
Dear reviewer,
please find attached the Word file with point-by-point response to all comments

Round 2
Reviewer 1 Report
Comments and Suggestions for Authors
No additional comments.